# Italian Version of the Pittsburgh Rehabilitation Participation Scale: Psychometric Analysis of Validity and Reliability

**DOI:** 10.3390/brainsci11050626

**Published:** 2021-05-13

**Authors:** Marco Iosa, Giovanni Galeoto, Daniela De Bartolo, Valentina Russo, Ilaria Ruotolo, Grazia Fernanda Spitoni, Irene Ciancarelli, Marco Tramontano, Gabriella Antonucci, Stefano Paolucci, Giovanni Morone

**Affiliations:** 1Department of Psychology, Sapienza University of Rome, 00185 Rome, Italy; daniela.debartolo@uniroma1.it (D.D.B.); valentina.russo@uniroma1.it (V.R.) gabriella.antonucci@uniroma1.it (G.A.); 2IRCCS Fondazione Santa Lucia, 00179 Rome, Italy; grazia.spitoni@uniroma1.it (G.F.S.); m.tramontano@hsantalucia.it (M.T.); s.paolucci@hsantalucia.it (S.P.); g.morone@hsantalucia.it (G.M.); 3Department of Human Neurosciences, Sapienza University of Rome, 00185 Rome Italy; giovanni.galeoto@uniroma1.it (G.G.); ilariaruotolo9696@gmail.com (I.R.); 4IRCCS Neuromed Pozzilli, 86077 Isernia, Italy; 5Department of Dynamic, Clinical Psychology and Health Studies, Sapienza University of Rome, 00185 Rome, Italy; 6Department of Life, Health and Environmental Sciences, University of L’Aquila, 67100 L’Aquila, Italy; irene.ciancarelli@univaq.it

**Keywords:** rehabilitation, neurorehabilitation, participation, compliance, stroke, psychometry, reliability, validity

## Abstract

Patient’s active participation in therapy is a key component of successful rehabilitation. In fact, low participation has been shown to be a prognostic factor of poor outcome; however, participation is rarely assessed in clinical settings. The Pittsburgh Rehabilitation Participation Scale (PRPS) is a validated, quick, and accurate measure of participation, relying on clinicians’ observation, and not requiring any self-report by patients. The aim of this study was to validate an Italian version of the PRPS. Following forward and back-translation of PRPS into Italian, the translated version was validated in a total of 640 therapy sessions, related to a cohort of 32 patients admitted to an Italian hospital. It was tested for concurrent validity, finding significant correlations with Barthel Index (*R* > 0.58, *p* < 0.001) and SF-36 Physical and Mental Health (*R* > 0.4, *p* < 0.02), for predictive validity, finding significant correlation with the effectiveness of rehabilitation (*R* = 0.358, *p* = 0.045), and for inter-rater and intra-rater reliability, computing an Intra-class correlation coefficient (ICC = 0.926 and 0.756, respectively). These psychometric properties results were similar to those of the original version of this scale. The proposed PRPS can be helpful for Italian clinicians in the assessment of patient’s participation during rehabilitation.

## 1. Introduction

Patient’s active participation to their own therapy has progressively been considered as a key point for a patient-centered treatment [1], and as a relevant factor influencing the efficacy of rehabilitation [2]. Patient’s motivation has been stated as the most important, although the most difficult, part of the work of the therapeutic professions [1]. In fact, participation is associated with the treatment outcome in terms of autonomy in activities of daily living (ADL) and of mobility [3]. Despite this interest about the role played by an active patient’s participation during therapy, only a few studies have assessed it [3,4,5,6,7,8].

Poor participation in rehabilitation could be due to many different factors: older age, depression, apathy, cognitive impairment, severe motor impairment, immobility syndrome, low self-efficacy, low confidence in the therapist’s ability to successfully rehabilitate, fatigue, comorbidities such as cardiac diseases, and personality factors [4,9].

Some studies measured the intensity of therapy in terms of total hours per day [10,11]. Although intensity is important, a crucial dimension is the quality of the interaction between patient and therapist for its potential role in rehabilitation, an aspect rarely evaluated. The quality of this interaction should be carefully assessed, especially in experimental rehabilitative programs, including those in which the patient–therapist interaction is mediated by a device related to newly emerging rehabilitation technologies [12,13].

Furthermore, patient’s participation is considered as a process that allows patients to be an integral part of the decisions and of the activities that influence their health [14]. In fact, patient participation is acknowledged as a key component of quality and effective rehabilitation, and it should be a core element of person-centered care, an approach focused on the patients’ individual needs, wants, and preferences [2].

The Pittsburgh Rehabilitation Participation Scale (PRPS) has been validated as the first published scale for rating patient participation [8]. This scale measures the participation of patients in their physical and occupational therapy. It is a very simple scale, with a single item having 6 possible scores: (1) no participation (the patient refused the entire session); (2) poor; (3) fair; (4) good; (5) very good; and (6) excellent participation (the patient participated in all exercises with maximal effort, finished all exercises, and actively took interest in exercises and/or future therapy sessions). The original version of this scale is reported in the Appendix A with detailed descriptions of the above six levels. This scale relies on clinicians’ observation, requiring no self-reported data from the patients [8]. The objective of this study was to validate an Italian version of the PRPS.

## 2. Materials and Methods

The validation of an Italian version of PRPS included two main parts: the translation procedure, and the evaluation of its psychometric properties compared with those of the original version of the scale [8]. The translating procedure [15,16] and that for evaluating psychometric properties were similar to those already performed for validating other clinical scales in previous studies [8,17,18].

The translation procedure included three steps. Firstly, two official translators, English native speakers, independently of each other, translated the original PRPS into Italian language (forward translation). This phase involved the joint work of a translator with a technical background and one with a medical background, the latter also judging the efficiency of the translation. Subsequently, two bilingual people, independently from each other and unaware of the original version, translated the Italian scale into English. These last two translations of the English-language scale were then independently re-translated into Italian by two health professionals with English language certification, unaware of the original version (backward translation). Lastly, all the translators gathered to decide the definitive translation of the PRPS Scale. With the purpose of adapting the translated scale to Italian culture, the translated scale was reanalyzed by a group of experts specializing in different medical disciplines. Experts had the opportunity to comment on elements of the translation by inserting their comments on a form. Once tested for validity and reliability, the translation judge examined this final version of cultural adaptation and approved it. The final Italian version of PRPS is reported in the Appendix A of this paper. Given its simplicity, neither particular difficulties nor controversies were recorded in the translation process described above.

For the validation process, a cohort of patients admitted in different complex operative neurorehabilitation units of an Italian hospital was recruited. Patients aged under 18 years were excluded, as were patients with severe cognitive disabilities, and non-native Italian speakers.

The Italian version of PRPS was administered to 32 patients during the first and the last ten sessions of physical therapy during their stay in a neurorehabilitation hospital (for a total of 640 therapies). Twenty-six of the 32 enrolled patients were further evaluated in their first session by a second therapist blind to the scoring of the first. This last procedure allowed for assessing the inter-rater reliability according to the procedure already employed in the original PRPS standardization [8]. As in that study, we also evaluated the inter-rater reliability by means of Two-Way Random, Single-Measure, Intra-class Correlation Coefficient (ICC(2,1)). The intra-rater reliability was assessed by a single therapist separately evaluating the 10 first and 10 last sessions of 32 patients by computing the One-Way Random, Single-Measure, Intra-class Correlation Coefficient (ICC(1,1)). Furthermore, the Cronbach’s alpha was also computed in our study.

Concurrent validity was measured by comparing the average PRPS assessed in the first and last ten sessions with the admission and discharge scores of the Barthel Index, respectively, and a 36-Item Short Form Survey (SF-36) by means of Spearman’s correlation coefficient (R). The validated Italian version of Barthel Index (BI) was used [15]. BI assesses the independency in the activities of daily living; it is formed of ten items and has a total score ranging from 0 (total dependency) to 100 (total independency). BI was preferred to the functional independence measure used in the original study [8], because BI is more common in neurorehabilitation and especially in Europe [18] (in particular in Italy: many regional healthcare systems adopt the BI as the main standard outcome). The validated Italian version of the SF-36 was used for the assessment of health-related quality of life, covering two main domains: physical and mental health [17].

Predictive validity was assessed by means of the correlation coefficient computed between the mean PRPS averaged among the first 10 sessions, with the effectiveness of rehabilitation evaluated as the percentage change in the scores of BI and SF-36 from admission to discharge, with respect to the maximum obtainable change [19]. Wilcoxon rank test was used to assess the change between admission and discharge scores. Finally, the responsiveness of PRPS was computed by assessing the probability of very good or excellent participation (scores 5 or 6) in the first/last session of therapy associated to the BI score at admission/discharge, respectively (BI score was categorized in groups of 10 points). A parameter logistic model, conventionally used in item response theory, was used for fitting the data by means of the least squares method. All statistical analyses were performed using the Statistical Package of Social Sciences (SPSS), version 23.0 for Windows.

The study was approved by the Independent Local Ethics Committee of the hospital in which data were collected, and all participants signed an informed consent form.

## 3. Results

The demographic and clinical characteristics of the enrolled sample are shown in Table 1. From admission to discharge, a significant wide improvement was observed for BI score (*p* < 0.001). Additionally, quality of life slightly, but significantly, improved (SF-36PH: *p* = 0.018 and SF-36MF: *p* = 0.005). Participation in rehabilitation assessed with the proposed Italian version of PRPS slightly but significantly varied between the first and the last session of therapy (*p* = 0.04). Conversely, the PRPS score was not significantly different between the first and the tenth session of therapy.

In the first session of rehabilitation, the two raters assigned the same scores in 77% of cases, with a difference of only one point in the other 23% of cases. The resulting inter-rater reliability for the Italian version of PRPS was statistically significant, with an ICC = 0.926 (Table 2). Additionally, the intra-rater reliability was statistically significant, with values of ICC > 0.7 both in the first and the last ten sessions of physical therapy (Table 2).

Regarding concurrent validity, the scores of the Italian version of PRPS were significantly correlated with the BI scores and SF scores at both admission and discharge (Table 2). Then, the scores of the Italian version of the PRPS assessed in the first 10 sessions were found to be significantly correlated with the effectiveness of therapy in terms of BI scores, but not of SF-36 scores (Table 2).

The frequency distribution of the PRPS scores is reported in Figure 1. The upper plot shows that in most of the sessions the scoring ranged from 4 to 6, both in the first and in the last sessions. The minimum score of PRPS (score = 1) was recorded in 1.2% of cases, related to 4 of the first sessions of a single patient. He had a stroke, and he was the most severely affected patient at admission (BI = 10) and the second oldest participant (age = 85 years). However, in the last sessions of therapy, his participation increased to a PRPS score of 5, and he was discharged with a BI score of 30. The lower plot of Figure 1 shows the distribution of PRPS scores with respect to the simultaneous assessment of BI scores (occurring at the first session at admission and at the last session at discharge). According to the paradigm of the item response theory, higher PRPS scores were found more frequently in patients with higher ability in terms of BI scores (as confirmed by a coefficient of determination of the Parameter Logistic Model of 0.815).

## 4. Discussion

The psychometric analysis of the proposed Italian version of the PRPS showed good results in terms of intra- and inter-rater reliability, concurrent validity, predictive validity, and responsiveness. As shown in Table 2, the psychometric properties of the Italian version of the PRPS were close to those found in the original study [8]. The correlation coefficients evaluated for assessing concurrent validity were even superior in our study, despite being not very high and less significant than those found by Lenze and colleagues [8] for the differences in sample sizes between the two studies. The predictive validity was confirmed by a statistically significant correlation between the average score of the PRPS assessed in the first 10 sessions of physical therapy and the effectiveness of therapy evaluated in terms of improvement in terms of Barthel Index score. This finding was in accordance with previous studies on the potential role of participation as a prognostic factor of rehabilitation outcome [2,4,6]. However, the predictive capacity was not significant in terms of SF-36 scores, neither for physical nor for mental health-related domain. However, it should be noted that patients showed a slight improvement in terms of quality of life (less than 10% of the maximum possible improvement), despite a gain in the independency in the activities of daily living of about 40% of the maximum achievable. The proposed Italian version of PRPS was able to predict this last high improvement.

We also found a significant improvement in the PRPS score from the first to the last sessions of rehabilitation, similar to that observed in the original study [8]. In general, the psychometric properties of the proposed Italian version of the PRPS were in line with those of the PRPS reported in the original study [8]. The inter-rater reliability between physical therapists found for the Italian version was 0.926; it was 0.96 in the original study [8]. The intra-rater reliability, not assessed for the original version of the PRPS, was statistically significant in our study, with values of ICC higher than 0.7 both at admission and discharge. The concurrent validity in our study was assessed by *R* > 0.58, when the mean PRPS was compared with BI scores at admission or discharge, whereas in the original study it was *R* = 0.38 when the mean PRPS was compared to the motor domain of the functional independence measure evaluated at admission [8]. The predictive validity in the original study was computed with the correlation between mean PRPS and change in functional independence measure finding an *R* = 0.32 [8]. We found a similar result when testing the correlation between the mean PRPS evaluated in the first ten sessions and the effectiveness in terms of the Barthel Index, which assesses the independency in the activities of daily living with a similar value of correlation: *R* = 0.358 [2,8].

Lenze and colleagues found a slight increment of PRPS in the first nine therapy sessions after admission (of about 0.4 points) [8]. We also found a slight improvement in PRPS score from admission to discharge (0.5 points), but in more than twenty sessions. In that study and ours, patients were elderly with neurological (about two-thirds) and orthopedic (about one-third) diseases. These conditions (aging and diseases) are known to often be associated with depression and cognitive difficulties: neurological damages impact physical, cognitive, psychological, and emotional functioning, as well as quality of life [20,21]. According to these observations, and also in our study, neither physical nor mental domains of quality of life showed a wide improvement. However, the mental health at admission was found to be significantly correlated to the participation in rehabilitation treatment.

On average, our PRPS score was slightly higher than that recorded in the original study [8] and increased less in the first nine sessions. It is noteworthy that more than 90% of sessions received a PRPS score equal to 4, 5 or 6. The first three scores were rarely recorded, but they were associated to severely affected patients. On one hand, this low sensitivity may reduce the use of this scale in clinical settings with less severely impaired patients; on the other hand, the sporadic refusal of therapy could be considered as an important alarm for clinicians because it may limit the efficacy of the entire rehabilitation.

Our study confirmed that patient’s participation is one of the most reliable predictors of rehabilitation outcome, and physical therapists should carefully take into account this issue in their clinical practice, when cognitive problems also subsist. The important role of participation in the rehabilitation process should be quantitatively assessed and increased because it may help to reduce long-term disability [2], according to the main aim of rehabilitation as defined by the World Health Organization [22].

The main aim of this study was to propose an Italian version of the PRPS, and we found psychometric properties similar to the original version of PRPS. However, the results of our study should be considered despite its limits. The most important limitation is the reduced size of the enrolled patient sample. It should be noted that the validity and reliability of PRPS was already proven in the original [8] and other [2,4] studies; we just tested the Italian version of this scale, although given our reduced sample size we did not test the PRPS scores among patients with different pathologies, as performed in the original study [8]. Another limitation could be that inter-rater reliability was tested on two therapists. We evaluated the intra-rater reliability: this measure was absent from the original study [8]. The number of evaluated sessions for each patient was higher in our study than in the original (20 vs. 9). Our study also showed a limitation of the scale, related to not-homogenous frequency distribution of its scores, with the upper scores more frequent than the lower ones.

The development of the PRPS and subsequent validations in different languages could be helpful for therapists and clinicians for taking into account the participation of patients in their own therapy sessions, a factor strictly correlated to functional outcomes. Furthermore, with the wide diffusions of innovative rehabilitation methods, especially those involving new technologies, it could be fundamental to assess the active participation of patients to these new approaches [12,13,23,24]. In fact, especially for elderly patients, some technologies could be alienating and generate anxiety, risking patients abandoning therapy [25]. However, cognitive factors could influence the outcome of technological rehabilitation [26]. In this scenario, it is also fundamental for clinical staff to understand the needs and expectations of patients for increasing their motivation during therapy [27]. Properly developing motivating tasks, and also exploiting new technologies, may increase the participation of subjects, therefore reducing perceived fatigue during therapy [28]. Another crucial aspect is that the participation of patients to therapy could depend on their relationship with the therapist. In fact, therapeutic relationships could vary depending on patient’s and therapist’s expected outcomes, on their interpersonal affective bonding, on their reactions to difficulties [29], and it could even vary over time [29]. This relationship may also depend on the patient’s trust in the therapist’s abilities, a factor that may improve active participation in therapy [30], impacting the attitudes and expectations of patients’ familiars and caregivers, and on the eventual presence of psychological support [31]. Diffusion of the PRPS, validated in different languages across the world, could provide a deeper insight in these aspects for improving therapeutic alliance among stakeholders, and hence rehabilitation outcomes.

## 5. Conclusions

The proposed Italian version of the Pittsburgh Rehabilitation Participation Scale (PRPS) showed concurrent validity, a predictive validity, and an inter-rater reliability similar to those found for the original version of this scale [8]. Additionally, the properties of intra-rater reliability and responsiveness were satisfied by the Italian version of PRPS. Despite the limitations of our study described above and those of this scale, which should be carefully taken into account, the ease of using the PRPS could allow for progressive assessment of patients’ participation in their own rehabilitation.

## Figures and Tables

**Figure 1 brainsci-11-00626-f001:**
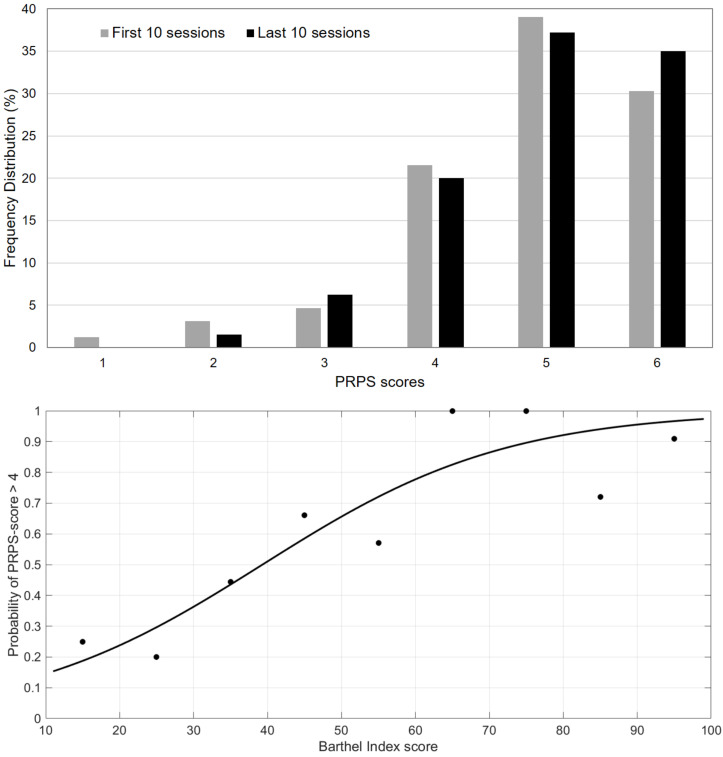
Frequency distribution of PRPS score. In the above plot are the distribution of scores in the first (grey bars) and last (black bars) ten sessions. In the below plot are the percentages of PRPS scores equal to 5 (very good) or 6 (excellent) in the first and last sessions associated to the BI score at admission and discharge, respectively (dots represent data, whereas the curve represents the fit obtained using a Parameter Logistic Model).

**Table 1 brainsci-11-00626-t001:** Mean ± standard deviation or percentages of demographical and clinical features.

Parameter	Description	Values
Age	Years	60.3 ± 18.0
Gender	Male	62.5%
	Female	37.5%
Pathology	Stroke	37.5%
	Other Neurological Disorders	28.1%
	Orthopedic Pathologies	34.4%
Barthel Index	Admission	59.9 ± 28.0
	Discharge	73.4 ± 25.5
	Effectiveness	39.4 ± 42.2%
SF-36 Physical Health	Admission	40.7 ± 7.3
	Discharge	43.8 ± 8.8
	Effectiveness	5.0 ± 13.0%
SF-36 Mental Health	Admission	41.1 ± 9.5
	Discharge	46.8 ± 10.3
	Effectiveness	8.6 ± 16.4%
PRPS	Therapist 1 First session	4.6 ± 1.3
	Therapist 2 First session	4.8 ± 1.1
	Therapist 1 First 10 sessions	4.8 ± 1.0
	Therapist 1 Last 10 sessions	5.0 ± 0.9
	Therapist 1 Last session	5.1 ± 0.8

**Table 2 brainsci-11-00626-t002:** Comparisons of the psychometric properties of the Italian version of PRPS and those found in the original study [8] (BI stands for Barthel Index, FIM for motor domain of the functional independence measure, SF-36 for Short Form, PH for physical health, MH for mental health, ICC is intra-class correlation coefficient, R stands for Pearson’s correlation coefficient in the original study and for Spearman correlation coefficient in our study, 10S stands for ten sessions of therapy).

Psychometric Properties of PRPS	Results of the Italian Version of PRPS	Results of the Original Study on PRPS [8]
Mean PRPS score	4.91 ± 1.03 (range: 1–6)	4.73 ± 0.76 (range not reported)
PRPS score increment	From 4.78 ± 1.24 to 4.87 ± 1.13 in 9 sessions (*p* = 0.47, Wilcoxon test) From to 4.78 ± 1.24 to 5.13 ± 0.79 in all sessions (*p* = 0.04, Wilcoxon test)	From 4.29 ± 0.93 to 4.67 ± 1.04 in 9 sessions (*p* < 0.0001, *t*-test)
Inter-rater reliability	ICC = 0.926 Cronbach’s alpha = 0.962	ICC = 0.91 for occupational therapists ICC = 0.96 for physical therapists
Intra-rater reliability	ICC = 0.844, Cronbach’s alpha = 0.982 (first 10S) ICC = 0.756, Cronbach’s alpha = 0.969 (last 10S)	Not assessed
Concurrent validity	*R* = 0.633 (*p* < 0.001) with BI *R* = −0.400 (*p* = 0.023) with age *R* = 0.518 (*p* = 0.002) with SF-36PH *R* = 0.433 (*p* = 0.013) with SF-36MH	*R* = 0.38 (*p* < 0.001) with FIM *R* = −0.21 (*p* < 0.001) with age
Predictive Validity	*R* = 0.358 (*p* = 0.045) with BI-effectiveness *R* = 0.222 (*p* = 0.222) with SF36PH-effectiveness *R* = 0.035 (*p* = 0.851) with SF36MH-effectiveness	*R* = 0.32 (*p* < 0.001) with change in FIM *R* = −0.13 (*p* < 0.05) with length of stay

## Data Availability

The data of this study are published in a private repository on Zenodo, and can be available upon reasonable request to the corresponding author.

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
