# Peer review of "Italian Version of the Pittsburgh Rehabilitation Participation Scale: Psychometric Analysis of Validity and Reliability"

_brainsci, 2021, doi:10.3390/brainsci11050626_

Round 1

Reviewer 1 Report

This is an important study, as participation in rehabilitation indeed correlates with outcome.

Introduction

I would like to see some more information about the PRPS in the introduction, for instance a description of the 6 levels. Moreover, the main validation properties of the PRPS could be given, as these are the standards with which the results of this study will be compared.

Measuring participation in rehabilitation is of importance for therapists to see whether they can improve the rate of participation of patients by improving their relation with the patient. This part of the value of the PRPS is not mentioned. Please reflect on this, including relevant literature, for instance following line 43. Also the impact of families is not mentioned. Although this is not the objective of this study, it still is important to list it as an important factor (see the original study of the PRPS).

Material and Methods

I would suggest to place the comment about the approval of the study (lines 66-68) to the end of this section.

Reading about the transaltion process, I was wondering what kind of difficulties and discussions had been occured. Maybe, one or two examples could be given, followed by a thorough description in a supplementary file.

In order to defend the choices for establishing the concurrent validity, the two scales (SF-36 and BI) should be described in more detail. Especially, the SF-36 has to be completed by the patient himself. Did the authors used an Italian (validated) version? I doubt about the sentence that these scales are the gold standars (line 102), without any references. Maybe, other scales are of greater importance. In the original study of the PRPS, the FIM was used. Why not choosing this scale? Please motivate. Furthermore, in  the result section two different scores of the SF-36 have been used. This should have been clarified in this part of the manuscript.

I think, it is not correct to mention that in this study the same techniques are used as in 'the original study' (i.e. line 104), as different scales have been used.

Results

The results are written in a telegram-like style. I would prefer to write more extensively. For instance, in line 123 and following, the results are presented about the INTERRATER RELIABILITY, showing the scores of the TWO raters. Both pieces of information are important to mention. This comment counts for all aspects.

I don't understand the sentence in lines 121-122 Als, the results in the section between lines 128-134ure not understandable to me. In both cases, the used English phrase is incorrect. What is meant with 'a good responsivity'?

Discussion

Much effort has been done to compare the results of this study to the resuls of the original study of the PRPS. I would suggest to present the figures in a table, so one can see in one view the differences and the similarities.

Tha authors state that the small group size is not a problem, beacuse 'it only is a translation'. However, in the original study, a difference was seen between stroke patients vs hip patients. Because of the small group size in this study, no subdivisions could be made, what is limiting the significance of the results.

Looking at the scores of the PRPS, one can wonder whether this scale is sensitive enough. Low scores are not given, as far as I understand (lowest mean score is 4.6, with 6 being the highest socre). The group difference of the mean score between the first and the last session is only halve a point (0.5). On an indvidual level, this is negligible and has no meaning. So, probably the PRPS can be used in a research project, but it seams unfit as a clinical tool. I would suggest, the authores reflect on this issue, especially related to the last part of the discussion (lines 203-215), where they suggest to use the PRPS in the clinical setting.

Reviewer 2 Report

I appreciate the opportunity to review the manuscript “Italian version of Pittsburgh Rehabilitation Participation Scale: psychometric analysis of validity and reliability”. This study examines the psychometric features of PRPS in a Italian sample. I appreciate the topic of this research, but I have some suggestions for improvement the quality of the paper:

  1. Table 3 and related results. Although it was found a significant correlations between average PRPS in the first 10 sessions and BI and SF-36 PH, the correlational coefficient was low (R=0.663 and R=0.518). Thus, I suggest to specify it in the results and conclusions sections of paper.
  2. In the text of the manuscript, there was no reference of figure 1. Please, argue it in the results section.
  3. The conclusions of the authors were more optimistic. I suggest to moderate them

Round 2

Reviewer 1 Report

I would thank the authors for their extensive responses and the changes they made to the manuscript. I appreciate their efforts in order to meet suggestions I made after the first version.

I have still three points to mention.

First, I think it is still needed to check the manuscript for correct use of the English language.

Secondly, unfortunately, Table 2 seems a mess. I see a lot of empty cells, and the cells that are filled with results are very narrow, resulting in a strange picture, that is not readible. So, I cannot comment on this addition.

Finally, the authors state in their cover letter, that they added a sentence about the sensitivity and the clinical value of the PRPS. This was the conversation:

Looking at the scores of the PRPS, one can wonder whether this scale is sensitive enough. Low scores are not given, as far as I understand (lowest mean score is 4.6, with 6 being the highest score). The group difference of the mean score between the first and the last session is only halve a point (0.5). On an indvidual level, this is negligible and has no meaning. So, probably the PRPS can be used in a research project, but it seams unfit as a clinical tool. I would suggest, the authores reflect on this issue, especially related to the last part of the discussion (lines 203-215), where they suggest to use the PRPS in the clinical setting.
AUTHORS: This is another very important aspect, and we would thank the reviewer to have highlighted. As explained above, we have now provided more details, that we think improved our work. About the discussion of the results already reported above we have written in Discussion the following new sentence: “In mean, our PRPS-score was slightly higher than that recorded in the original study [8], and varied slower in the first 9 sessions. It is noteworthy that more than 90% of sessions received a PRPS-score equal to 4, 5 or 6. The first three scores were rarely recorded, but they resulted associated to severely affected patients. On one hand this low sensitivity may reduce its application in clinical settings in which patients are less severely impaired, on the other hand the sporadic refuse of therapy could be considered as an important alarm for clinicians working with severely affected patients because it may limit the efficacy of the entire rehabilitation.”

Unfortunately, I cannot find this addition in the edited manuscript. Please add yet.
